# An Asymptomatic Patient with Fatal Infertility Carried a Swedish Strain of *Chlamydia trachomatis* with Additional Deletion in The Plasmid *orf1* that Belonged to A Different MLST Sequence Type

**DOI:** 10.3390/microorganisms7070187

**Published:** 2019-06-28

**Authors:** Valentina A. Feodorova, Sergey S. Zaitsev, Yury V. Saltykov, Edgar S. Sultanakhmedov, Andrew L. Bakulev, Sergey S. Ulyanov, Vladimir L. Motin

**Affiliations:** 1Laboratory for Molecular Biology and NanoBiotechnology, Federal Research Center for Virology and Microbiology, Branch in Saratov, 410028 Saratov, Russia; 2Department for Skin Diseases, Saratov State Medical University named after Razumovsky, 410028 Saratov, Russia; 3Department for Medical Optics, Saratov State University, 410012 Saratov, Russia; 4Departments of Pathology and Microbiology & Immunology, University of Texas Medical Branch, Galveston, TX 77555-1019, USA

**Keywords:** *Chlamydia trachomatis*, new variant, nvCT, ddCT, MLST, asymptomatic genital Chlamydial infection, deletion, cryptic plasmid

## Abstract

Here, we present the first case of asymptomatic genital Chlamydial infection caused by the new emerging *Chlamydia trachomatis* (*C.t.*) ST13 strain genovar E, which has a double deletion of 377 bp and 17 bp in *orf1* gene of the cryptic plasmid (ddCT). This case occurred in an infertile patient (case-patient) with a detectable level of Chlamydial antibodies and a spermatozoa deficiency known as azoospermia. Additionally, the ddCT strain showed the presence of a duplication of 44 bp in the plasmid *orf3* and SNP in *orf4*, which were known as the typical characteristics of the Swedish variant of *C.t.* (nvCT) genovar E. Multilocus sequence typing (MLST) determined a significant difference between ddCT and nvCT in four alleles (*oppA, hfiX, gitA and enoA*). Both ddCT and nvCT were assigned to different genetic lineages and could be allocated to two different non-overlapping clonal complexes. Furthermore, ddCT demonstrated a considerable difference among 4–5 alleles in comparison with other *C.t.* strains of genovar E of ST4, ST8, ST12, and ST94, including the founder of a single relevant cluster, wtCT E/SW3 (Swedish genetic lineage). In contrast to other genovar E strains, ddCT had identical alleles with seven out of seven loci found in ST13 strains of genovars D and G, including the founder for this clonal group, D/UW-3/CX, and six out of seven loci found in its derivatives, such as ST6, ST10, and ST95 of genovars G and H. Nevertheless, MSTree V2 showed that ddCT and nvCT could have a common early ancestor, which is a parental *C.t.* G/9301 strain of ST9. A significant difference between ddCT and nvCT of genovar D (nvCT-D) that was recently found in Mexico was also determined as: (i) ddCT belonged to genovar E but not to genovar D; (ii) ddCT had a 44 bp duplication within the *orf3* of the plasmid typical for nvCT; (iii) ddCT possessed an additional 17 bp deletion in the *orf1*. In conclusion, improved case management should include the clinical physician’s awareness of the need to enhance molecular screening of asymptomatic Chlamydia patients. Such molecular diagnostics might be essential to significantly reducing the global burden of Chlamydial infection on international public health.

## 1. Subject

Here, we report a case of asymptomatic genital Chlamydial infection caused by the *Chlamydia trachomatis* (*C.t.*) strain with double deletion in the *orf1* gene of the cryptic plasmid (ddCT) identified in the infertile patient (case-patient) with a detectable level of Chlamydial antibodies (the ddCT sequence reported in this research was deposited in GenBank, the accession number is MH595818.1). The ddCT strain had an exact 377 bp deletion in the *orf1* as the Swedish *C.t.* isolates known as the novel variant of *C. trachomatis* (nvCT). Similarly to nvCT, the ddCT belonged to the genovar E as determined by sequencing of the *ompA* gene. Nevertheless, in contrast to nvCT bearing only a single deletion of 377-bp in the *orf1* gene, ddCT additionally had a second upstream deletion of 17-bp in size within the same gene.

The main goal of the current study was to characterize the exact differences between the ddCT and nvCT variants. We applied multilocus sequence typing (MLST) to determine the sequence type (ST) for both ddCT and nvCT. We found an evident difference in polymorphism of their housekeeping genes, indicating that they belong to different sequence types.

The patient has given written informed consent to participate in the study and to publish the case details.

## 2. Case 

The case-patient, a man between 20–25 years old, sought care from the Clinic for Skin and Venereal Diseases, Saratov State Medical University, Saratov, Russia. The reason for this was the presence of a detectable level of Chlamydial antibodies observed in his serum (Table 1). He began engaging in sexual intercourse at the age of 16. Since that time, he had a single regular heterosexual partner from Saratov city. He indicated that he smokes. He denied any other pernicious habits or addiction to alcohol or any other previous Sexually Transmitted Diseases (STDs). He had no complaints, as well as clinical manifestations of any infections including bladder- or kidney-related infections. Physical examination showed no signs of an inflammation of the urethra. Palpation of the urethra and scrotum was painless and revealed no changes, including any swelling of the testicles. After the massage, there were urethral secretions and discharge from the penis. Per rectum examination demonstrated that the prostate gland was not enlarged, and that it possessed densely-elastic consistency, was painless, and the median furrow was clearly contoured. The seminal vesicles were not palpable. Ultrasound investigation on the pelvic region registered no pathology. However, specific Chlamydial immunoglobulin G (IgG) was confidently registered in ELISA in diagnostic titers in the case-patient serum (Table 1). The main sperm characteristics of the case-patient were the following: motility (immotile spermatozoa was 53% versus 50% of the lower WHO reference limit [1]) and vitality (35% versus 58% [1]) was changed progressively (Appendix A). Also, a mild spermatozoa deficiency such as azoospermia (sperm concentration was less 15 × 10^6^ spermatozoa per ml [1]) was registered. In order to examine the case-patient more closely, he was sent to the Federal Research Center for Virology and Microbiology, Branch in Saratov. Here, his individual clinical specimens were carefully tested by nucleic acid amplification tests (NAATs) for the detection of both plasmid and chromosomal targets of the *C. trachomatis* strains and the presence of both wild (wtCT) and nvCT variants.

Initially, two commercial NAATs were compared with respect to their sensitivities and specificities with the reference to international commercially available PCR kits [2]. The end-point PCR (PCR-Ep) (Research Institute of Epidemiology, Moscow, Russia, AmpliSens *Chlamydia trachomatis*-Eph), which aimed to detect the plasmid DNA of the wtCT, found no Chlamydial DNA in the genital specimen (urethra) of the case-patient (Table 1). In contrast, a dual-target real-time PCR (PCR-RT) assay (Real-Best DNA *Chlamydia trachomatis*, Vector-Best, Novosibirsk, Russia) targeting both the cryptic plasmid and the *gyrA* gene with the same specimen revealed a positive response indicating the presence of *C.t.* DNA (Table 1). 

Furthermore, the presence of genetic material of nvCT in the specimen of the case-patient was confidently confirmed in PCR with a panel of primers targeting different parts of the *C.t.* cryptic plasmid. PCRs analysis detected the 377 bp deletion in *orf1*, a 44 bp duplication in *orf3*, and a Single Nucleotide Polymorphism (SNP) in *orf4*, all known to be characteristic of nvCT [3] and, with the exception of the duplication in *orf3*, were all found in the new variant of the nvCT (nvCT-D) that was identified recently in Mexico [4]. Therefore, the obtained data suggested the presence in our specimen of the nvCT variant that went undetected by using the PCR-Ep kit targeting the plasmid gene *pgp3 (orf3)*. 

The presence in ddCT of a 377 bp deletion typical of the *orf1* in the cryptic plasmid of both nvCT and nvCT-D strains was confirmed by sequencing the PCR fragment obtained with the primers flanking this deletion [5]. Moreover, sequencing the upstream region to the 377 bp deletion of the *orf1* revealed an additional 17-bp deletion (Table 1, Figure 1).

The primers amplifying this broader area of the *orf1* were described previously [6,7]. To distinguish between the double deletion strain of *C.t.* detected in the current study in the case-patient and nvCT and nvCT-D, we designated our *C.t.* variant as ddCT. The presence of a cryptic plasmid in this ddCT was also confirmed by a positive PCR result with primers detecting the *orf8* gene (Table 1). The *ompA*-based genotyping was performed by both amplification and sequencing of the relevant Variable Domain (VDII) of the *ompA* gene [8] and the 1156 bp-fragment of VDI–VDIV of the *ompA* gene as we described recently [9]. Both strains, ddCT and nvCT, had identical *ompA* regions with no SNP identified. Phylogenetic reconstruction of the *ompA* sequences of ddCT and nvCT compared with the *C.t.* reference strains [9,10] using the UMPGA hierarchical clustering method in MEGA 7 [11] showed that both strains could be seen in a single cluster of the *C.t.* genovar E variants on the dendrogram (Figure 2a). In contrast, nvCT-D belonging to genovar D [4] was assigned to another cluster corresponding to the genovar D strains (Figure 2a).

Surprisingly, multilocus sequence typing (MLST) based on seven housekeeping genes (*gatA, oppA, hfiX, gitA, enoA, hemN* and *fumC*) [9,12,13,14] determined that ddCT and nvCT had a difference in four alleles, such as *oppA, hfiX, gitA* and *enoA*, and belonged to ST13 and ST4, respectively (Table 2). Recently, when the relevant concatenated sequences of ST13 and ST4 were aligned and analyzed in MEGA 7 [11], they have been assigned to different genetic lineages of Group I and Group III as founders for each of the Groups [9,13,14]. Reconstruction of the genetic relationships of the allelic profiles of ddCT queried against MLST sequences of nvCT and the *C.t.* reference strains from the PubMLST database (Available online: https://pubmlst.org) in MSTree V2 [15] demonstrated that ddCT and nvCT could be allocated to two different non-overlapping clonal complexes (Figure 2b).

Overall, ddCT demonstrated a strong difference with no less than 4–5 alleles compared with other *C.t.* strains of genovar E, belonging to ST4, ST8, ST12 and ST94 that form a single cluster with the founder wtCT E/SW3 strain (Swedish genetic lineage) (Figure 2b). In contrast, ddCT had identical alleles with seven out of seven loci (100%) found in ST13 strains of genovars D and G, including the founder of this clonal group, wtCT, D/UW-3/CX strain, and six out of the seven loci (85.7%) found in its derivatives, such as ST6, ST10, and ST95 of genovars G and H (Table 2), and the early ancestor or parental *C.t.* G/9301 strain of ST9 (Figure 2b).

In summary, based on the MLST typing results, ddCT might be a new variant of *C.t.* genovar E of the ST13 cluster, bearing a 377 bp deletion in the *C.t.* cryptic plasmid that is typical for both nvCT and nvCT-D. 

## 3. Background

*C. trachomatis* is recognized as one of the most successful pathogens and has significantly over performed all other infectious agents in the number of cases of human genital infection among sexually active adolescents and young adults [16,17]. Chlamydial infection remains among the top sexually transmitted diseases (STDs) as the most commonly reported nationally notifiable disease for many countries [16,17]. In fact, despite the presence and availability of a number of commercial highly sensitive diagnostic NAATs, many cases of Chlamydial genital infection are still not detected [16]. The underlying reason of such inefficient diagnostics is related to the absence of specific clinical manifestations or an asymptomatic course for human genital Chlamydial infection. Such latent Chlamydia patients have no reasons to be seeking care and can be diagnosed accidentally during examination for the complications of Chlamydial infection [18]. The progression of undiagnosed infection can lead to severe reproductive complications in Chlamydia patients, such as pelvic inflammatory disease and tubal infertility [19,20]. PCR-based diagnostics has been developed and successfully applied widely for the detection of typical wild strains of *C. trachomatis* (wtCT) in human clinical specimens. The recent emergence of *C. trachomatis* strains known as a new Swedish variant (nvCT), resulted in a new wave of Chlamydial outbreaks at the beginning of 21st century [20]. These nvCT strains were defective in the *orf1* gene of the cryptic plasmid and were not detected by the PCR kits targeted to the relevant gene [6,7,21]. This diagnostic escape provided such strains with a great advantage over wtCT and led to their rapid spread in a number of European countries [6,7] and even in some regions of the Russian Federation [2,22,23]. In fact, after the first case of nvCT documented in Sweden in 2006, the proportion of these CT strains was dramatically increased, reaching up to 20–65% in some Scandinavian countries [21,24]. Retrospective analysis revealed the presence of nvCT in these regions since 2003. While being undetectable, nvCT could have produced at least 15,000 false-negative tests in Chlamydia patients during 2003–2006 [25]. Efficacy of laboratory diagnostics of *C.t.* was significantly improved when dual-target NAATs for the simultaneous detection of both wtCT and nvCT were included in a standard laboratory protocol of *C.t.* screening in patients suspicious for Chlamydia infection. As a result, the nvCT proportion markedly decreased in the counties implementing these diagnostic systems, falling from 56% in 2007 to 6.5% in 2015 [21]. In fact, only isolated cases of Chlamydia genital infection caused by nvCT have been reported worldwide recently [2,4,22,23,26,27]. However, it is clear that the reports of such nvCT-caused cases in Chlamydia patients could be relatively rare because the majority of countries only use NAATs for the detection of wtCT [1,9,17,20,27,28]. This illustrates the importance of including the dual-target NAATs detection methods for nvCT strains in the investigation of Chlamydia cases with false-negative results in PCR targeted exclusively at the wtCT strains [23,27]. Importantly, Chlamydia infection with nvCT was more frequently asymptomatic, suggesting a possible difference in virulence between the nvCT strain and the wtCT strain [18,23,27]. Nevertheless, generally no major genetic diversity was found between genomes of the nvCT and the wtCT strains detected in Sweden. Therefore, it is still unclear whether there is a difference in biological fitness between the two variants, which demonstrated high similarity with respect to epidemiological distribution and minimal differences in clinical signs in the vast majority of Chlamydia patients [29]. All Swedish nvCT strains tested thus far had identical MLST profiles. Thus, the transmission studies of nvCT in Sweden conducted for the past ten years suggested that Swedish nvCT variant is clonal and genetically stable [21]. However, a recent report described a new *C.t.* variant in Chlamydia patients, which also contained an identical deletion of 377 bp in the plasmid *orf1* [4]. This novel strain was detected in Mexico and showed a distinct polymorphism of the *ompA* and *pmpH* genes. In contrast to the Swedish nvCT strains belonging to genovar E [2,18,21,23,24,26,27,29], the Mexican *C.t.* strain was identified as genovar D (nvCT-D), and showed the absence of a 44 bp perfect tandem duplication within the plasmid *orf3* characteristic of nvCT [4].

Here, this is the first report on the existence of another nvCT-type variant, ddCT, containing additional upstream deletion in the plasmid *orf1* and belonging to different serovar E ST cluster.

## 4. Discussion

*C. trachomatis* infections, which most frequently are asymptomatic, still are a major public health concern globally [30]. Therefore, the updated European standards have aimed to improve the tools for testing and develop of clear recommendations to identify, verify, and report *C.t.* variants to better control their spread. Based on molecular epidemiology typing techniques, such as MLST, tools for cluster, network and phylogenetic analyses [31] were developed. These approaches are currently widely used to reveal transmission networks, risk groups, and evolutionary pathways resulting in the discovery of new variants of *C.t.*, nvCT and nvCT-D [4,6,7,31]. In the reported case, the *C.t.* strain detected by us and designated as ddCT was significantly different from the earlier identified Swedish nvCT due to two main characteristics: (i) ddCT had ST13 rather than ST4 characteristic as in all previously described genovar E nvCT strains (Table 2, Figure 2b); (ii) ddCT contained an additional 17 bp deletion within the *orf1* of the plasmid (Table 1, Figure 1). Our data revealed a notable difference of ddCT from other *C.t.* variants of ST4 of the genovar E strains, including nvCT (Sweden2), as well as with the founder for this clonal group, E/SW3 (wtCT) typical for Sweden and other European countries including Russia [13,14,27,32,33]. The ddCT reported here was strongly grouped in a ST13 cluster that originated from *C.t.* strains belonging to genovars that are different from E, such as G/9301 and D/UW-3/CX, which were identified early in genital sites of Chlamydia patients in Seattle, USA [32,34]. Thus, based on the MLST data analysis, ddCT and nvCT were assigned to different genetic lineages of *C.t.*. Nevertheless, both *C.t.* variants diverged from a common *C.t.* progenitor (Figure 2a,b). Moreover, a strong difference exists between ddCT and recently described Mexican nvCT-D in several characteristics: (i) similarly to nvCT, ddCT belonged to genovar E [2,18,21,23,24,26,27,29], while nvCT-D was identified as genovar D [4]; (ii) ddCT showed the presence of a 44 bp perfect tandem duplication within the *orf3* of the cryptic plasmid, which is typical of genovar E nvCT, whereas nvCT-D lacked this duplication; (iii) a unique additional 17 bp deletion was identified for the first time in the *orf1* of the ddCT plasmid (Table 1, Figure 1). Unfortunately, the data on the sequence type of the Mexican isolate nvCT-D were not presented for further comparison [4].

The appearance of additional deletion within the *orf1* already damaged by the lack of 377 bp fragment in this gene characteristic to Swedish strains was not very surprising, as the pseudogenes have a tendency to further accumulate the mutations. Nevertheless, this novel 17 bp deletion in ddCT can serve as a prominent marker for this lineage of the nvCT-type strains. Similarly, we predict further identification of *C.t.* variants with additional deletion(s) in the *orf3* of nvCT and ddCT due to the impairment of this gene by the 44 bp duplication, resulting in the appearance of another pseudogene on the plasmid. Overall, the most likely scenario for the formation of nvCT, ddCT, and nvCT-D was their divergence from a common ancestor strain by first obtaining a 377 bp deletion in the *orf1*, then dividing into genovars E and D, followed by gaining a 44 bp duplication in the *orf3* of the genovar E variant, and its further separation into ST4 (nvCT) and ST13 (ddCT). Another possible scenario is an independent formation of identical 377 bp deletion and 44 bp duplication in *C.t.*s with different genovars and sequence types. This scenario is less likely as there are no direct or inverted repeats present in the *orf1* and *orf3*, which could serve as points for recombination.

The most important question which still remains is the role of plasmid or its *orf1* in *C.t.* pathogenesis [6,7,35]. The *orf1* encodes a putative integrase of 305 amino acid (a.a.) residues in size, and a 377 bp deletion in nvCT strains, while removing a primer binding site for the diagnostic NAATs can theoretically lead to the formation of a shortened but still active 178 bp peptide [35]. In this respect, the presence of an additional 17 bp impairment upstream of 377 bp deletion in ddCT will ensure the integrase inactivation. Then, the apparent toxic effect on the case-patient spermatozoa can indirectly indicate a certain relationship between the main sperm characteristics, especially motility and vitality (Appendix A) and the integrase activity. This hypothesis requires additional studies.

## 5. Conclusions

The number of genital tract *C.t.* infections is steadily increasing worldwide with approximately 50–70% of infections being asymptomatic and causing a significant burden on health care systems [38]. Despite the significant progress in molecular diagnostics of wtCT, as well as our knowledge of *Chlamydia* evolution, this study demonstrates the presence in Chlamydia patients of a novel nvCT variant with diverse modifications in the loci that could be used as diagnostic and genotyping targets. This type of strains could compromise certain NAATs, resulting in a hidden emerging epidemic. Therefore, improved case management should include clinical physician’s awareness of the need to enhance molecular screening of asymptomatic Chlamydia patients. Such molecular diagnostics might be essential to significantly reducing the global burden of Chlamydial infection on international public health.

## Figures and Tables

**Figure 1 microorganisms-07-00187-f001:**
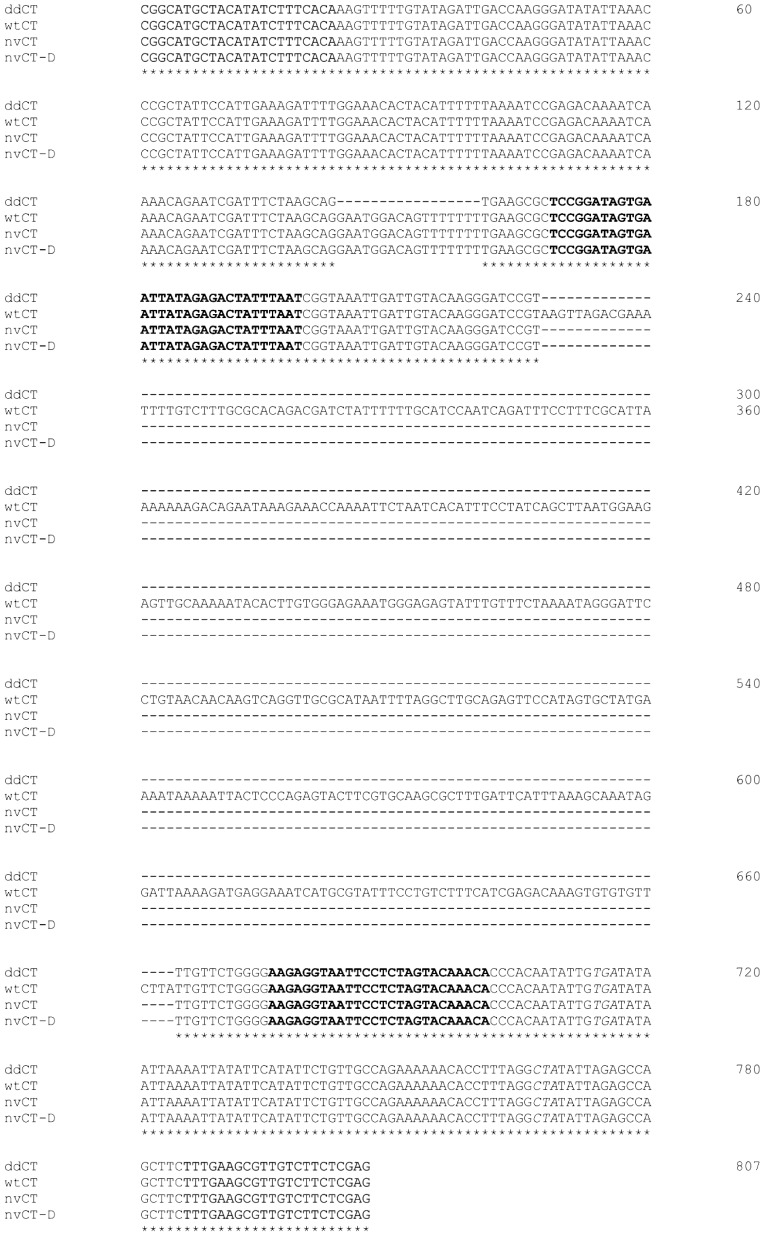
Alignment of the *orf1* of the cryptic plasmid of *C. trachomatis* for wtCT, nvCT and nvCT-D with the partial corresponding sequence of the case-patient sample from urethra (ddCT) amplified with primers gB_swCT-F/R (highlighted). Stop codons for the *orf1* (TGA) and *orf2* genes (TAG, in complementary chain) are in italic. The primers swCT_serE-F/R flanking the 377 bp deletion in the nvCT strain are in bold. The sequence of the case-patient contains both 17-bp and 377-bp deletions.

**Figure 2 microorganisms-07-00187-f002:**
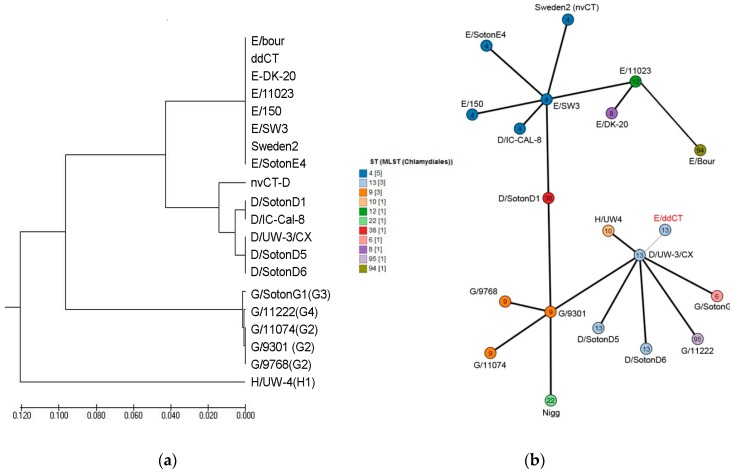
(**a**) Phylogenetic tree demonstrating the relationship of the *ompA* sequences of ddCT (GenBank accession No. MH782166), nvCT, nvCT-D (GenBank accession No. KY474386) to reference strains of *C.t.* genovar E from Table 2. Thick branches, subtending all large clades, represent 100% bootstrap support. The scale at the bottom represents one substitution per site. (**b**) GrapeTree clustering of 10 STs available in *Chlamydiales* PubMLST database (Available online: https://pubmlst.org/chlamydiales/). Each node corresponds to a single ST with an individual color (see on the left) represented by the relevant *C.t.* genovar(s)/clinical isolate (indicated around each node). Two *C.t.* deletion variants, ddCT and nvCT (Sweden2) belong to different clusters, ST13 and ST4, respectively.

**Table 1 microorganisms-07-00187-t001:** Detection of *Chlamydia* from the case-patient.

The case-Patient Gender	Age	Clinical Sample from the Site	PCR-nvCT	PCR-orf2 wtCT/nvCT	Duplication 44 bp in orf3 of nvCT	SNP in orf4 of nvCT ^1^	PCR-orf8	Genovar ^2^	PCR-Ep ^3^	PCR-RT ^4^	DIFT	Chlamydial IgG Titers	Presence of the Deletion(s) in *orf1* of the Cryptic Plasmid	
													Typical for nvCT of 377-kb	Additional of 17-kb
Male	23	Urethra	+	−/+	+	+	+	E	−	+	+	1:8	+	+

^1^ Single nucleotide polymorphism detected in the *ompA* gene: the SNP position compared to the genovar E, strain E/Bour (GeneBank accession no. HE603212); ^2^
*C. trachomatis* genovar was determined based on sequence of the VDII-VDIV regions of the *ompA* gene; ^3^ End-point PCR (AmpliSens *Chlamydia trachomatis*-Eph) targeting plasmid DNA of the wtCT; ^4^ A dual-target real-time PCR (PCR-RT) assay (Real-Best DNA *Chlamydia trachomatis*) detecting both the cryptic plasmid and the *gyrA* gene.

**Table 2 microorganisms-07-00187-t002:** The *ompA* and seven housekeeping genes MLST sequence types of ddCT, nvCT and reference strains of *C. trachomatis* obtained from the PubMLST database (Available online: https://pubmlst.org).

ST (MLST)	Allele							*ompA* genovar(s)	Reference Strain(s) Information		
	*oppA*	*hfiX*	*gitA*	*enoA*	*hemN*	*fumC*	*gatA*		Name (Accession Number of Identical Sequences in PubMLST Database if Available)	*ompA* genovar	Reference
4	3	1	1	2	4	2	3	E	E/Sweden2 (269)	E	[29,36]
	3	1	1	2	4	2	3		E/SW3 (236)		[32]
	3	1	1	2	4	2	3		E/SotonE4 (237)		[32]
	3	1	1	2	4	2	3		E/150 (234)		[32,34]
	3	1	1	2	4	2	3	D	D/IC-Cal-8 (7)	D	[13]
6	3	3	2	5	3	1	3	G	G/SotonG1 (246)	G	[32]
8	2	1	1	2	4	2	3	E	E/DK-20 (5)	E	[13]
9	3	3	2	4	3	2	3	G	G/11074 (242)	G	[32,34]
	3	3	2	4	3	2	3		G/9301 (244)		
	3	3	2	4	3	2	3		G/9768 (245)		
10	3	3	2	1	3	2	3	H	H/UW-4 (18)	H	[13,28]
12	3	4	1	2	4	2	3	E	E/11023 (220)	E	[32,34]
**13**	**3**	**3**	**2**	**5**	**3**	**2**	**3**	**E**	**E/Saratov-2u, ddCT**	**E**	**This study**
	3	3	2	5	3	2	3	D	D/UW-3/CX (215)	D	[32,34]
	3	3	2	5	3	2	3	D	D/SotonD5 (232)	D	[32]
	3	3	2	5	3	2	3	D	D/SotonD6 (233)	D	[32]
	3	3	2	5	3	2	3	G1/G2	N/A	G	[14]
22	8	11	9	11	11	7	8	N/A	Nigg (45)	N/A	[37]
38	3	1	2	2	4	2	3	D	D/SotonD1 (231)	D	[32]
94	3	4	1	32	4	2	3	E	E/Bour (235)	E	[28,32]
95	3	3	28	5	3	2	3	G	G/11222 (243)	G	[32,34]

^1^ N/A, Not applicable.

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
