# Peer review of "An Asymptomatic Patient with Fatal Infertility Carried a Swedish Strain of Chlamydia trachomatis with Additional Deletion in The Plasmid orf1 that Belonged to A Different MLST Sequence Type"

_microorganisms, 2019, doi:10.3390/microorganisms7070187_

Round 1
Reviewer 1 Report
In this case report, Feodorova et al., describe the identification of a new and distinct Chlamydia trachomatisnisolate from a patient displaying C. trachomatis specific antigens in his serum. PCR based analysis of the cryptic plasmid revealed that the plasmid contained the same signatures as the cryptic plasmid from the Swedish C. t isolate nvCT. Interestingly, the cryptic plasmid was also found to harbor an additional deletion in the orf1 gene. MLST analysis suggests that this variant is a genovar E isolate and that its lineage is distinct from the Swedish isolate which is also a genovar E isolate. Importantly, this study highlights the importance of using a multiple target PCR based assay for diagnostic detection of Chlamydia in patients, given that detection using only one locus can result in false negatives, like in the case of failure to detect the nvCT variants since detection primers were targeted to a region in the plasmid orf1 gene which is missing in the nvCT variant.
Major concern:
Overall, I find that this study is missing whole genome analysis. There is now increasing evidence that omp based genotyping in many cases does not correctly identify serovars, and I think that MLST might have similar limitations. Whole genome sequencing (WGS) provides a much more robust method for genovar/lineage and cluster classification with respect to other C. t clinical variants. WGS can also detect presence of novel hybrids resulting from inter-genovar recombination events and can detect chromosomal and plasmid-based SNPs, indels, and rearrangements, that could shed light into virulence mechanisms that might drive patient observed pathologies. In this regard, I find this to be a major flaw in the study and I would definitively like to see this kind of analysis being incorporated in all case reports. WGS can be routinely done in a relatively inexpensive manner given the small sizes of Chlamydia genomes. It shouldn’t be a limiting factor. Furthermore, the catalogue of fully sequenced C.t clinical isolates continues to increase, providing valuable information available for the highest level of resolution when doing phylogenetic analysis of new isolates. I would encourage the authors to strongly consider this in future work.
However, given that the scope of a case report might not necessarily require this level of analysis, the results provided in this work is sufficient for publication. Methods, results, and conclusions are relevant, and substantiated, and provides the field with information of a new clinical isolate. English and nomenclature needs some work prior for work to be accepted for publication. Please find minor comments in this regard below.
Minor edits:
Line 48: 377 bp deletion in the orf1 gene as the Swedish ….
Line 48 and throughout manuscript: Change CT to C.t
Line 52: size within the same gene.
Line 89: gyrA should be italicized
Line 93: Replace “These were positive PCRs for detection of” with “PCR analysis detected the”
Line 94: and a Single Nucleotide Polymorphism
Line 95: Replace “except” with “with the exception of”
Lines 96 and 97: Gramar needs to be revised
Line 102: In Table 1 some of the column headers cannot be read.
Line 108: orf1 should be italicized.
Line 110: insert the word “genes” after “orf2”.
Line 116: insert word “gene” after “orf8”.
Line 133: Replace the word “clasters” with “clusters”.
Line 159: Insert “and” before “has significantly over..”
Line 166 and throughout manuscript: Chlamydia should be italicized and in some cases the first letter has to be capitalized. This is a common occurrence throughout the manuscript and a major breach of nomenclature convention.
Line 168: “undetected timely” grammar should be revised.
Line 255: Authors claim that orf1 might possibly participate in a host-pathogen interaction is completely baseless without having any information on the genome of this strain. This statement should be removed. There is insufficient data to even speculate about this.
Author Response
Dear Sir/Madam,
We would like to thank both reviewers for the valuable suggestions to improve the quality of the manuscript. Please, find below a detailed response to the reviewer’s comments.
We strongly agree with the reviewer that the whole genome analysis would be quite beneficial for further characterization of this C. trachomatis variant. Nevertheless, we would like to note that this work was done using clinical sample, therefore, whole genome sequencing is still a challenge for such samples. We are working on obtaining C. trachomatis isolate from this specimen to conduct a whole-genome sequencing.
All minor edits have been corrected through the test.
Reviewer 2 Report
Manuscript details:
Journal: Microorganisms
Manuscript ID: microorganisms-505589
Type of manuscript: Case Report
Title: Asymptomatic patient with fatal infertility carried Swedish strain of Chlamydia trachomatis with additional deletion in the plasmid orf1 and belonged to a different MLST sequence type
Authors: Valentina Feodorova *, Sergey Zaitsev, Yury Saltykov, Edgar Sultanakhmedov, AndrewL. Bakulev Bakulev, Sergey Ulyanov, Vladimir Motin * Submitted to section: Medical Microbiology, https://www.mdpi.com/journal/microorganisms/sections/medical_microbiology
Chlamydiae and Chlamydia like Bacteria
https://www.mdpi.com/journal/microorganisms/special_issues/chlamydia_2019
Dear Editor,
The authors of this manuscript describe a new variant of Chlamydia trachomatis genotype E, with the deletion of 377 bp in the previously described nvCT and an additional 17 bp deletion in the orf1 gene of the cryptic plasmid (named double deletion C. trachomatis, ddCT) in an asymptomatic patient with fatal infertility. The molecular study is well performed, although epidemiological transcendence and its implication in the infection management should be the main orientation of the text. In my opinion, the description of this new strain is very interesting, but the text is too long (~2900 words). Althought Microorganisms has no restrictions on the length of manuscripts, this Case Report Paper should be provided more concise (abstract <200 words, text <1800 words) for better understanding. I suggest the Editor “Reconsider after Major Revisions” and I encourage the authors to rewrite the paper following some general suggestions to adapt the text to a typical Case Report form.
Kind regards,
Luis Piñeiro
Author Response
Dear Sir/Madam,
Although the manuscript may look lengthy than a typical Case Report, we believe that it should contain all necessary details related to the molecular characterization of novel C. trachomatis variant. The lack of these details will neglect the importance of the existence of different types of Swedish variants. Taking into account that the original Swedish variant had gone undetected for a number of years, we want to be sure that there will be no future cases of missed diagnostics of these types of strains.
Round 2
Reviewer 2 Report
Unfortunately, the authors have not followed nor responded in detail my suggestions. The text is too long and with redundancies that must be avoided for better understand by the readers, without remove the details needed. So, I encourage the authors to read once more my comments, make the necessary changes and answer each suggestion if they agree or not.
Best regards.
Author Response
Dear Sir/Madam,
Thank you for providing us a detailed guidance for improving the manuscript. In response to your request, we revised the manuscript within the proposed space limit. Particularly, we reduced the abstract up to 228 words, and the body test up to 2,127 words as recommended.